# What sets aeolian dune height?

Andrew Gunn [1,2,3], Giampietro Casasanta [4], Luca Di Liberto [4], Federico Falcini[5], Nicholas Lancaster [6] & Douglas J. Jerolmack [3,7✉]

Wherever a loose bed of sand is subject to sufficiently strong winds, aeolian dunes form at predictable wavelengths and growth rates. As dunes mature and coarsen, however, their growth trajectories become more idiosyncratic; nonlinear effects, sediment supply, wind variability and geologic constraints become increasingly relevant, resulting in complex and history-dependent dune amalgamations. Here we examine a fundamental question: do aeolian dunes stop growing and, if so, what determines their ultimate size? Earth's major sand seas are populated by giant sand dunes, evolved over tens of thousands of years. We perform a global analysis of the topography of these giant dunes, and their associated atmospheric forcings and geologic constraints, and we perform numerical experiments to gain insight on temporal evolution of dune growth. We find no evidence of a previously proposed limit to dune size by atmospheric boundary layer height. Rather, our findings indicate that dunes may grow indefinitely in principle; but growth depends on morphology, slows with increasing size, and may ultimately be limited by sand supply.

[1] School of Earth Amtosphere and Environment, Monash University, Clayton, Australia. [2] Department of Geological Sciences, Stanford University, Palo Alto, USA. [3] Department of Earth and Environmental Sciences, University of Pennsylvania, Philadelphia, USA. [4] Institute of Atmospheric Sciences and Climate - National Research Council of Italy (CNR-ISAC), Rome, Italy. [5] Institute of Marine Science - National Research Council of Italy (CNR-ISMAR), Rome, Italy. [6] Earth & Ecosystem Sciences, Desert Research Institute, Reno, USA. [7] Department of Mechanical Engineering and Applied Mechanics, University of Pennsylvania, Philadelphia, USA. ✉email: sediment@sas.upenn.edu

Earth's major sand seas are often populated with giant dunes, up to hundreds of meters in height and kilometers in wavelength. These massive sediment piles, visible from space on our planet and across the Solar System, indicate that conditions for sand transport have persisted for millenia. Unraveling how giant dunes form therefore has implications for understanding atmospheric flows and climatic stability. The initial wavelength and growth rate of aeolian dunes from a flat sand bed are well understood; aerodynamic theory developed for idealized conditions has recently been extended and successfully applied to predict dune formation in nature[1–3]. Once dunes grow large enough to perturb the flow nonlinearly, however, size regulation becomes more complicated. Dunes calve and merge through collisions and wake interactions[4,5]; but the net effect is pattern coarsening through time[6–9]. Is there any limit to the size that aeolian dunes can grow, besides time? One elegant hypothesis is that the size of giant dunes is limited by the averaged mixed layer height (MLH), where a stable resonance condition is found between topographic and capping-layer waves[7]. This prediction is appealing because it suggests a general and physical (rather than site specific and geological) control by atmospheric forcing, and that the scale of giant dunes can be used to infer the MLH on other planets[10]. An alternative hypothesis, however, is that dune growth just slows logarithmically with time, as dunes grow larger and their migration rates diminish[6]. As real dune fields evolve over century and longer timescales, additional site-specific boundary conditions have been suggested to exert a control on dune size[11,12]: sediment supply, geologic constraints, wind variability, and climatic stability. Neither the MLH control, or the logarithmic slowing hypothesis, have been directly tested in nature.

## Results

**Observations**. Global LANDSAT imagery was used to manually identify and delineate the boundaries of 38 dune fields (Methods). We utilized ERA5 reanalysis data to determine 10-m hourly wind velocity for the 2008–2017 decade on a nominally 32-km horizontal grid[13]. Potential sand flux ($\vec{q}$) was estimated from these data with a linear excess stress model that explicitly incorporates an entrainment threshold[14,15] (Methods); it is important to note that this corresponds to the saturated sand flux, and true flux could be less if supply is limited. We utilized SRTM ASTER GDEM V3 topography to determine the average dune geometry —wavelength, $x$, height, $z$ and width, $y$—within each $32^2$-km$^2$ tile[16] (Figs. 1 and S1; Methods); topographic resolution prohibits detection of dunes with $x < 100$ m. Corresponding dune morphology was manually categorized into the canonical types; barchanoid, transverse, linear, and star[15,17,18]. Taken together, our analysis produces estimates of modern sand flux, and dune geometry and morphology, for 2,093 $32^2$-km$^2$ tiles on Earth. Where possible, we used published data to estimate dune-field age (Methods). Mixed layer height was determined using all available daytime CALIPSO satellite measurements collected from 2006 to 2019 over each dune field (Methods). These are always collected in the early afternoon, where the boundary layer is convective and most likely to promote sand transport[19], but there is still a clearly identifiable delineation between the aerosol-laden mixed layer at the free-atmosphere above[20].

We first examine patterns in dune geometry for the global dataset. Although previous studies have documented self-similar scaling of barchan dune geometry[21], those observations did not include other dune geometries or giant dunes. Our compiled data show that dune geometry is not self similar for the largest wavelengths, where very high aspect-ratio dunes are observed (Fig. 2a). Plotting width against wavelength produces distinct

clouds of data that correspond to dune morphology; barchanoid and star dunes follow a 1:1 line, while linear dunes are the widest and transverse dunes show intermediate behavior (Fig. 2d). Another distinction is that the highest dunes in the dataset ($z > 100$ m) are disproportionately represented by star dunes, which also appear to only form at large wavelengths[22,23] (generally > 1 km) (Fig. 2a). In contrast, aspect-ratio scaling for barchanoid and transverse dunes generally follows observed patterns for subaqueous dunes[24,25].

It is well established that dune morphology is a consequence of variability in wind direction: predominantly unidirectional sand flux results in barchanoid and transverse dunes under conditions of relatively low and high sand supply, respectively; oblique and bi-directional sand flux creates linear dunes; and highly variable sand flux directionality gives rise to star dunes[15,17,18,26,27]. How this variability influences dune geometry and ultimate size, however, has not been fully examined. We compute a flux directionality measure as the ratio of the magnitude of the resultant potential sand flux vector ($|\sum \vec{q}|$) over the absolute potential sand flux ($\sum |\vec{q}|$) that varies from 0 associated with net-zero flux, to 1 corresponding to unidirectional flux (Fig. 1f). This is similar to the ratio of resultant to absolute so-called 'drift potential'[18,28] (i.e., RDP/DP). Perhaps unexpectedly, ostensibly unidirectional barchanoid and transverse dunes exhibit a wide

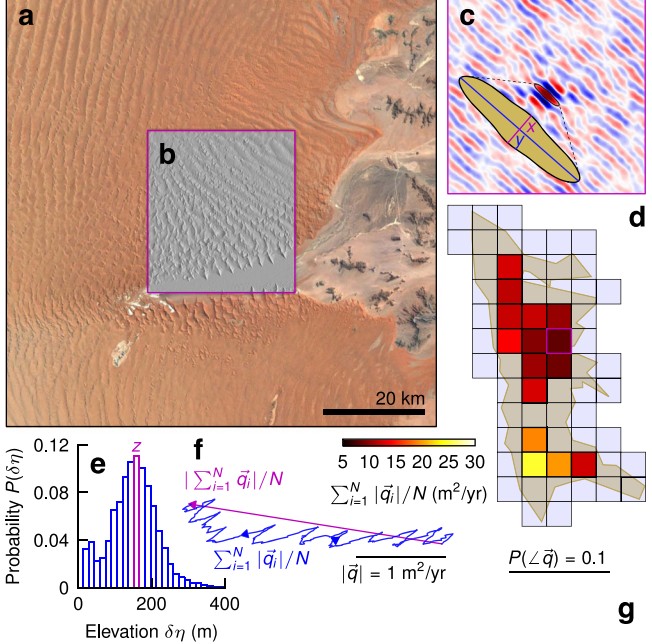

**Fig. 1 Extraction of dune geometry and sand flux. a** LANDSAT imagery of part of the Namib Sand Sea, one dune field in the dataset. **b** Hillshade SRTM topography from an example $32^2$-km$^2$ tile. **c** The high-pass autocorrelation of the topography in **b** overlaid by the extracted characteristic planform dune geometry in black (zoomed inset in yellow defines wavelength $x$ in magenta and width $y$ in blue). **d** Grid of prospective tiles intersecting the dune field (yellow); tiles included in the dataset (where dune geometry can be measured) are colored by mean sand flux $|\vec{q}|$ inferred from ERA5 10-m winds. **e** Probability distribution of local relief $\delta\eta$ found by convolution of SRTM topography with a min-max box of width $x$; the peak marks the characteristic dune height $z$. **f** Time-means of the resultant sand flux vector (magenta) and cumulative sand flux vectors (blue) for **b**; terms denote their lengths, and arrows their directions. **g** The probability distribution of sand flux directions for **b**. Black lines denote scale in **a**, **f** & **g**, $N$ is the number of hourly measurements over the decade of ERA5 reanalysis, north is up in **a**–**d**, **f** & **g**, and magenta boxes in **b**–**d** outline the common tile.

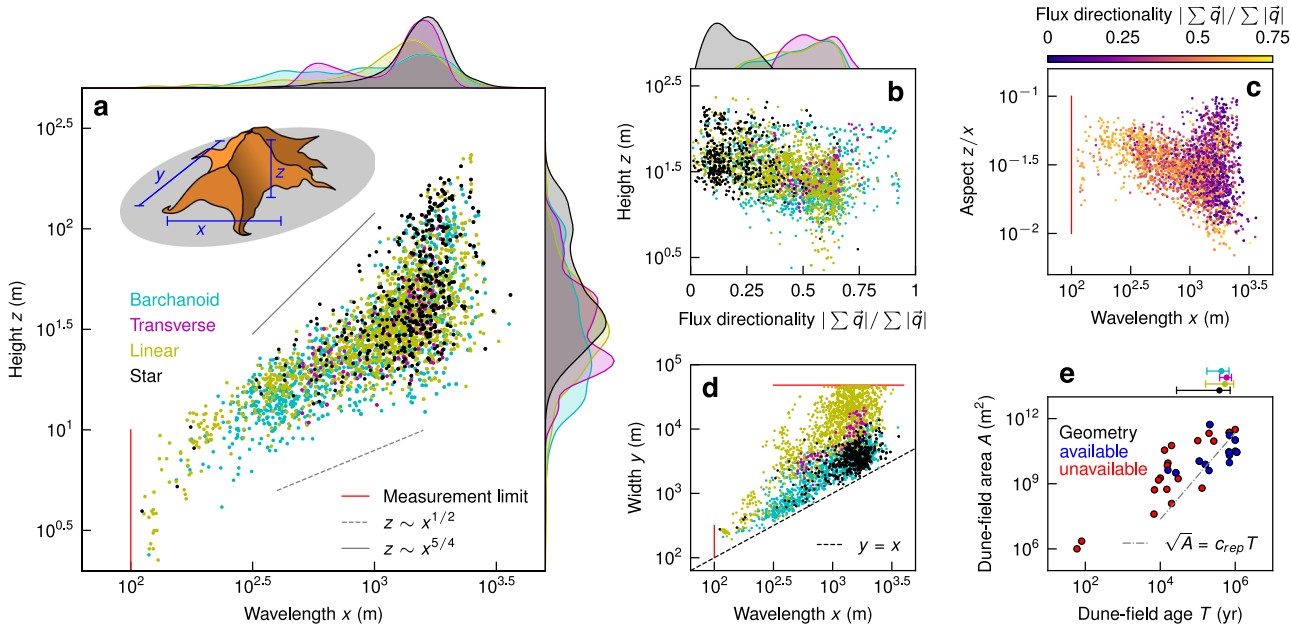

**Fig. 2 Trends in Earth's aeolian dunes. a** Characteristic dune wavelength $x$ and height $z$ for 2,093 $32^2$-km$^2$ tiles. Points and kernel density estimates for each axis colored by type (barchanoid, cyan; transverse, magenta; linear, yellow; star, black), power-laws bounding the distribution given in gray, and a schematic defining $x$, $y$ and $z$ for an example star dune in upper left. **b** Flux directionality (i.e., the resultant sand flux magnitude over the sand flux magnitude sum, or purple over blue in Fig. 1f) against dune height $z$. Points and kernel density estimate colors defined in **a**. **c** Dune wavelength $x$ against aspect $z/x$, points colored by flux directionality using the colorbar above. **d** Wavelength $x$ against width $y$ colored as in **a**. The dashed black line marks $y = x$, by definition points lie above this line. **e** Dune-field age $T$ against area $A$ for 29 dune fields with a powerlaw $\sqrt{A} = c_{rep}T$ (dot-dashed gray line), where $c_{rep} = 0.48$ (m/yr) is a representative dune migration rate. Blue points ($n = 11$) are included in the geometric analysis, red are not. Using the blue points and sharing the age-axis, dune-type ages (mean ± standard deviation) are given above the parametric plot. Red lines in (**a**, **c**, **d**) mark measurement limits.

range of values for flux directionality[29] (Fig. 2b). We attribute this noise to many potential factors, but of high significance are: first, sand flux directionality is determined over only 10 years — a relatively short time compared to the age of large dunes in the database — and therefore may not represent formative conditions; and second, sand supply is an important but unmeasured control on sand flux that likely varies significantly across dune fields. Star dunes, however, correspond only to low directionality (high variability) conditions as expected (Fig. 2b). The compiled data also reveal a previously unobserved trend: dune height is inversely related to flux directionality; i.e., dunes with low directionality are relatively taller (Figs. 2b and S2). Indeed, the previously discussed trend of decreasing aspect ratio with increasing wavelength is associated with more undirectional sand-flux regimes, while at the largest wavelengths, the cloud of points which buck this trend and have larger-than-expected aspect ratios correspond to lower flux directionality (Fig. 2c). These observations suggest that highly variable winds act to "pile up" sand, while more unidirectional winds create lower dunes.

We now turn our attention to the dune-field mixed layer height, and its potential control on the size of giant dunes. Although there are seasonal fluctuations in MLH, and variations among dune fields (Fig. 3), the averaged midday MLH $H$ varies little ($1 < H < 2$ km). Most importantly, we find no correlation between MLH and dune wavelength (Fig. 3b). In other words, data do not support the proposed control of MLH on limiting dune size[7]; in fact, dune wavelength exceeds MLH for most dune fields. To understand why, we must consider the proposed mechanism in light of the atmospheric conditions that give rise to sand transport. The MLH hypothesis assumed that the mixed layer is neutrally stable such that the interface between it and the free-atmosphere at $H$ is a capping interface; in this scenario, large dunes that perturb the flow can excite waves at the interface,

which then limit dune wavelength through a resonance condition[7]. While stability conditions that permit this behavior may sometimes occur, our analysis suggests that these conditions are not associated with sand transport. Rather, winds exceeding threshold are typically associated with strong instability[19]; the convection-enhanced mixing that enhances surface wind strength also destroys wave propagation, inhibiting resonance when sand transport occurs (see Text S1 for details).

While our observations are the most comprehensive to date, they still represent only a snapshot in time of the dune coarsening process. Factors important for the evolution of large dunes over millenia, such as sand supply and past variations in wind climate, are completely unconstrained. Further, the central question of what sets aeolian dune height remains unanswered. To access the trajectory of dune growth through time, and isolate and control boundary conditions that influence dune dynamics, we turn to numerical experiments.

**Numerical experiments.** We perform a suite of numerical experiments using ReSCAL[30], a model that couples cellular automaton rules for sediment transport with a lattice gas method for turbulent wind[30]. ReSCAL has been shown to produce many salient aspects of aeolian dune dynamics and morphology[8,30,31], and can be quantitatively scaled to nature[30]. Given that the history and boundary conditions of dune fields examined here are not known, however, we do not attempt a quantitative comparison of model runs with field data. Instead, we perform six numerical experiments that essentially bracket the range of Earth's aeolian landscapes[18]. Model runs conserve sand in a domain that is horizontally periodic. Domain height is set to be sufficiently large that it does not influence dune growth, informed by the lack of MLH control shown previously (Methods).

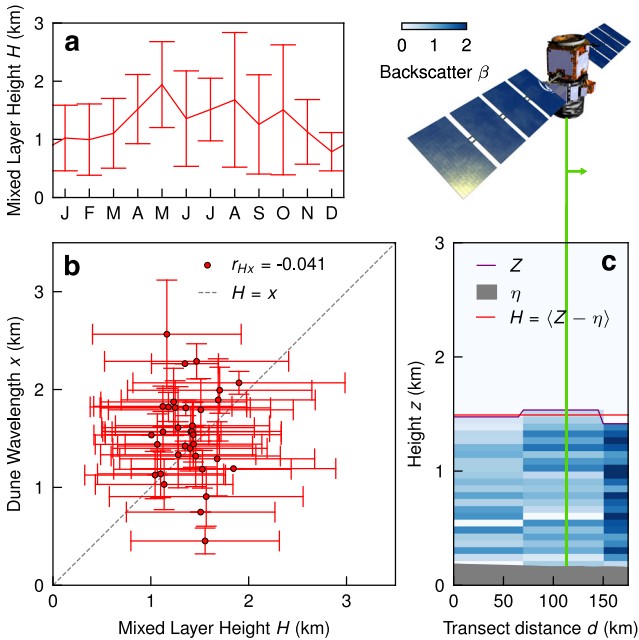

**Fig. 3 Mixed layer heights over dune fields. a** An example mixed layer height $H$ annual climatology for the Rub Al Khali measured using CALIPSO for 2006–2019. Monthly means and standard deviations given ($n = 222$). **b** $H$ and measured dune wavelengths $x$ for 34 dune fields in the geometry data set, means (red dots with black outlines) and standard deviations (red lines) for both measurements are shown, as is the Pearson's correlation coefficient $r_{Hx}$ and the identity $H = x$. If two characteristic dunes are identified in a tile, only the larger one is included in the averaging for this plot. **c** An example of the $H$ extraction from CALIPSO (pictured) over the Rub Al Khali. As the satellite passes over the dune field (gray region), the CALIPSO (green line) scan of the atmosphere detects high backscatter $\beta$ from aerosols in the mixed layer relative to the free atmosphere above (blue map, 5-km horizontal resolution). The mean difference (red line) of the delineation between high and low $\beta$, $Z$, (purple line) and elevation $\eta$ (gray region) for the scan constitutes one $H$ value[20].

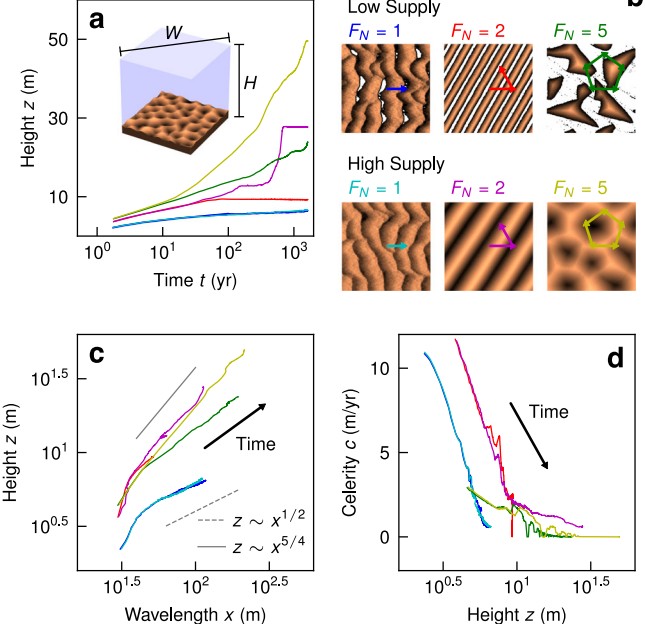

**Fig. 4 Numerical experiments of dune growth. a** Dune height time-series for ReSCAL experiments. Line colors correspond to experiments shown in **b**, a snapshot of the yellow experiment at $t = 162$ yrs shown to define the horizontally-periodic domain; $W = H = 522$ m). **b** Planform snapshots of each experiment at the final timestep $t = 1,624$ yrs; color is normalized elevation (dark is lower), white is non-erodible bedrock. The number of flux directions $F_N$ is given, as are the flux vectors for each experiment. The top row of low-supply experiments have $\eta = 3.5$ m of flat sand initially, whereas the bottom row have $\eta = 35$ m of flat sand initially. **c** Wavelength $x$ against height $z$ for each experiment coarsening over time; bounding powerlaws from the natural data (values in legend) given in Fig. 2a also shown. **d** Dune height $z$ against celerity (i.e., migration speed) $c$. Time arrows given in **c** and **d** for clarity.

The initial conditions are flat sand beds of two thicknesses, $\eta(t = 0) = 3.5$ m and $\eta(t = 0) = 35$ m, to simulate sediment-starved and sediment-saturated systems, respectively[4]. Three forcing regimes are chosen to mimic winds that produce unidirectional (barchanoid and transverse), linear, and star dune types by varying the number of wind directions $F_N$; these dune types correspond to flux directionality values of 1, 0.5 and 0, respectively. For $F_N > 1$, directions iterate every 4 months and all experiments are run for over 1,600 years. We verify that the imposed wind forcing produces the expected dune morphologies at the end of the model runs (Fig. 4b).

Each experiment shows that dune height grows approximately logarithmically with time, i.e., $z \sim \log(t)$ (Fig. 4a) as observed in previous dune simulations[6]. Deviations from this behavior are observed for linear dunes, as a result of dislocation repulsion[5,32]. Systems with high sand supply tend to produce dunes that grow taller when flux is not unidirectional (Fig. 4a), following intuition. Unidirectional dunes exhibit sub-linear scaling of height with wavelength indicating a decrease in aspect ratio as dunes grow. Star and linear dunes, by contrast, show super-linear $z − x$ scaling; their height grows more rapidly than unidirectional dunes, and they are relatively taller for all wavelengths (Fig. 4c). These qualitative behaviors are in accord with our observations from natural dune fields (Figs. 2a and 4c). For all conditions, numerical experiments show that dune migration rate (commonly called celerity) slows as dunes grow larger; while this behavior is a

well known consequence of mass conservation[8,15,33,34], higher-order effects like slip-face development and flow shielding may also reduce flux and hence migration rate as dunes become large[6,29,35]. Notably, star dunes become essentially stationary once their height reaches $\approx 10$ m due to their net-zero flux.

ReSCAL is subject to uncertainty in the conversion of time and length scales from the virtual to real domain (Methods), and the model omits secondary flows in the wind created by topography[30]—which may be particularly important for linear and star dunes[4,23]. Nevertheless, numerical experiments reproduce the main geometric and morphological patterns observed in natural dune fields and laboratory experiments[4,11,17,26,27,34], giving us some confidence that the temporal dynamics of dune growth in the model have some bearing on natural sand seas. In the absence of MLH control, modeled dunes coarsen indefinitely, but their growth rate slows over time, under constant forcing.

## Discussion

The distilled interpretation of our findings is this: Earth's giant dunes are growing ever-slower with size, and are not limited in size by MLH generically. This calls into question planetary studies that use the capping layer hypothesis to estimate MLH from observed dune wavelength[10]. Nevertheless, the presence of dune fields still places a strong constraint on atmospheric dynamics: near-surface winds must regularly exceed the entrainment threshold, but not by much, in order to maintain saltation that grows dunes[36]. With rudimentary knowledge of the composition

of the atmosphere and the sand grains, the dune-forming wind conditions on other planets may be determined with reasonable confidence[37].

Returning to our findings, snapshots of mature dunes in the numerical experiments (taken at $T \gtrsim 500$ yrs) are similar in geometry and morphology to the large dunes populating Earth's surface today. Estimating dune age using available measurements (Methods), we see the four morphologies of dunes have similar mean ages; if anything, star dunes are slightly younger than other large dunes (Table S1, Fig 2e). We conclude that Earth's star and linear dunes, with low flux directionality, are taller because they grow faster; reversing winds act to pile up sand. The numerical experiments also explain other details in the observed data: dune aspect is more sensitive to sediment supply in low flux directionality systems (Figs. 4a and 2c), and ever-slowing coarsening produces the negative skew of dune size probability distributions (Fig. 4a).

But these conclusions leave us with a conundrum: why are there no dunes for $x \gtrsim 2$ km, if they always grow? Coarsening rates for such large dunes are exceedingly slow. Over the millenia required to evolve dunes of this size, we hypothesize that climatic and geologic constraints become limiting. First of all, climate must remain sufficiently arid and windy for dunes to remain unvegetated and active; this becomes increasingly unlikely for timescales longer than the Holocene, i.e., $10^4$ yr[38,39]. Second of all, sand supply becomes increasingly likely to limit dune growth, as dunes pile sand higher and scour deeper into the substrate; many of the world's giant dunes show signs of sand limitation such as bare non-erodible interdune surfaces[18,23]. While perhaps neither satisfying nor surprising[4,11,12,40], our findings suggest that both the size and morphology of Earth's largest dunes are the integrated product of the unique geology and climatic history of each dune field. Nevertheless, universal trends in aeolian dune geometry, and the new relations observed between geometry and morphology, may be used to understand where observed dunes sit in their respective growth trajectories alongside other metrics such as defect density[41].

Our results also contribute to understanding the size of aeolian dune fields themselves[42]. Although scattered, we observe a positive trend in dune-field age ($T$) against area ($A$) (Fig. 2e), which could imply that dune-field expansion is driven by dune migration[19,28]. To test this idea, we utilize a representative upper bound on dune migration speeds from the numerical experiments: $c_{rep}$, the mean celerity after $t > 500$ yr for all six experiments (Fig. 4d). A first-order advective growth scaling can be anticipated, $\sqrt{A} = c_{rep}T$. The data follow the scaling, which indicates that at least some component of dune-field boundary expansion may be driven by dune migration itself. On the other hand, most dune fields lie above the scaling line, indicating they are larger than implied from expansion by dune migration alone; if true, this would suggest that dune-field size is set by properties of the sand supply[12]. It seems likely that flux directionality plays some role; in strongly unidirectional cases like White Sands, boundary expansion is clearly related to dune migration[33,43], but for stationary fields of star dunes like the southeast Grand Erg Oriental[23], sand supply must be the dominant factor.

These findings serve as a springboard for investigating how, and how fast, dunes respond to transient forcing. In particular, how will aeolian landscapes adjust to changing climate, and how does their maturity and history influence this change? We see two features of our data that suggest that dunes can be sluggish relative to changing winds. First is the observation of superimposed dunes, with morphologies that are distinct from the larger dunes they ride on[22]. This implies that changing wind may not reorient the entire dune, but rather initiate the formation of new (and much smaller) dunes that slowly cannibalize the underlying larger dune as they grow — as observed for fluvial dunes in response to rapid changes in flow[44,45]. Second is the unexpectedly large variance in flux directionality for ostensibly unidirectional dunes (Fig. 2a), which indicates that many large dunes may have been sculpted by wind conditions that are different from those of the last decade. A rate-and-state framework where dune form, rather than scale, is the measure of landscape adjustment may be useful for understanding dune-field evolution and anticipating dune responses to climate change[46].

## Methods

**Dune-field ages & areas.** Dune-field age estimates are found from a literature review[38,47–69] and summarized in Table S2. These data are a subset of the INQUA Dune Atlas. Methods of estimation are from geochemical and optical dating techniques of the sediments beneath dune fields, aeolian accumulation rates and deposit thicknesses, and aerial imagery. Uncertainty in each age is subject to a variety of inconsistent processes and is reported differently across the data aggregation. Dune-field areas are found simply by tracing the dune-field extent in Google Earth using LANDSAT imagery, also provided in Table S2.

**Sand flux from ERA5 reanalysis.** A time-series of 87,672 hourly 10-m winds $\overrightarrow{U_{10}}$ (m/s) from 2008 to 2017 inclusive are transformed into approximate sand flux $\overrightarrow{q}$ (m$^2$/s) using a standard and consistent approach using threshold friction velocity. Friction velocity, $u_*$, is calculated as $u_* = |\overrightarrow{U_{10}}| \kappa / \log(10/z_0)$, where $\kappa = 0.41$ is von Karman's constant and $z_0 = 10^{-3}$ m is the roughness length at the scale of sand transport[43]. Next a threshold friction velocity is defined as $u_{*,cr} = \sqrt{g d \rho_s / \rho_f}/10$, where $g = 9.81$ m/s$^2$ is gravity acceleration, $d = 300 \mu$m is grain diameter, $\rho_s = 2650$ kg/m$^3$ is sand density and $\rho_f = 1.225$ kg/m$^3$ is fluid (air) density, giving $u_{*,cr} = 0.252$ m/s as a representative value[15]. Finally sand flux magnitude is defined as $\overrightarrow{q} = \{\angle \overrightarrow{U_{10}}, 25\rho_f/\rho_s \sqrt{d/g}(u_*^2 - u_{*,cr}^2)\}$ for $u_* > u_{*,cr}$ and $\overrightarrow{q} = \{$NaN$, 0\}$ otherwise[14]. In lieu of grain-size data for all locations, we chose constants for this calculation that are representative for Earth and not specific to any particular dune field.

**Dune geometry extraction.** Planform dune geometry is found through the following algorithm designed to automate the extraction of the characteristic dune dimensions: width, length, and height. This process is done for tiles of topographic data, where the tiling is set by the gridding of the wind data such that each characteristic dune geometry found has corresponding saturated ERA5-derived sand flux data calculated at the tile center. The tiles analyzed are those that have a majority of their area shared with a dune-field area (Methods) and meet the following criteria: they do not contain ocean or non-dune relief, and are not exclusively unpatterned sand sheets. These constraints leave us with 2093 tiles to extract dune geometry from. Below we first explain the algorithm precisely, then explain how it maps onto a physical definition of dune dimensions.

First, an auto-correlation $R_\eta$ of a 32$^2$-km$^2$ tile of ASTER topography $\eta(\lambda, \phi)$ (where $\lambda$ is longitude and $\phi$ is latitude) is created using FFT (blue-red shaded fields in Supplementary Fig. S1f&n). The two largest modes are omitted so that any broad, non-dune slopes in the topography do not impact dune-pattern identification; and the square tile is masked by a circle so that dune width is not biased by orientation. We take specific level-sets $\partial\Omega_\alpha = \{(R_\lambda, R_\phi) | R_\eta = \alpha, \Omega_\alpha \ni (0,0)\}$ for $0 < \alpha < \max\{R_\eta\}$ of $R_\eta(R_\lambda, R_\phi)$ that bound the origin as shapes which represent the planform dune geometry (green-yellow lines in Supplementary Fig. S1f&n). Taking $\partial\Omega_0$ is a poor level-set since patterns are complex and include dislocations (upper example in Supplementary Fig. S1g). Instead, we identify the appropriate level-sets by finding one or two local maxima in a plot of $\alpha$ against $\chi = A(\Omega_\alpha)/A(\text{conv}(\Omega_\alpha))$, the ratio of level-set area $A(\Omega_\alpha) = \iint \Omega_\alpha dR_\lambda dR_\phi$ over its convex hull area $A(\text{conv}(\Omega_\alpha)) = \iint \text{conv}(\Omega_\alpha) dR_\lambda dR_\phi$ (black lines in Supplementary Fig. S1h&p). We take the only, or two largest $A(\Omega_\alpha)$, maxima, excluding trivial maxima where $A(\Omega_\alpha) > (1 - \varepsilon) A(\text{conv}(\Omega_\alpha))$ or $A(\Omega_\alpha) \ll A(\text{conv}(\Omega_\alpha))$, as the planform shape of dunes in the tile (cyan points in Supplementary Fig. S1p). This is unless there is no local maxima because $\chi(\alpha)$ decays monotonically, in which case we found $\chi(\alpha) = 1.1$ as the representative level-set (cyan point in Supplementary Fig. S1h). Overall this method is robust and general for all tiles and allows extraction of the sole dune type, or both dune types if one is superimposed on the other, in the tile. The level-set is then converted from longitude-latitude coordinates to local meters and finally dune wavelength $x_{auto}$ and width $y_{auto}$ are defined as its short- and long-axes, respectively.

Dune height is then extracted afterward by first convolving a min-max box of width $x_{auto}$ (in lon-lat) across $\eta(\lambda, \phi)$, which gives a map $\delta\eta(\lambda, \phi)$ where each point has the value of the local range in $\eta$ within $x_{auto}/2$ in $\lambda$ or $\phi$ (Supplementary Fig. S1b,c,j&k). The peak of a histogram of this elevation range map $\delta\eta$ is defined as the characteristic dune height $z_{auto}$ (cyan in Supplementary Fig. S1a&i).

After automatic calculation of all tiles, planform and vertical dimensions were then calibrated against a random subset ($n = 25$) of manually extracted geometries

using ImageJ with a linear scaling such that $x/x_{auto} = y/y_{auto} = 1.51$ and $z/z_{auto} = 0.85$. This method is outlined graphically for two illustrative examples in Supplementary Fig. S1 and processed geometry data are available in the Source Data file.

This process is specifically aimed at identifying the dimensions of the constituent and representative dune—or small-scale dune and large-scale dune superposition—for each tile. Planform dune dimensions of 'wavelength' and 'width' are named as such to follow standard nomenclature, but strictly these are just the short- and long-axis dimensions of the extracted characterstic planform dune shape, respectively. The short-axis dimension 'wavelength' has been defined the past as the distance between crests or troughs, or the shortest distance across the erodible bedform where the non-erodible inter-dune surface is exposed[18,22,23,41]. Here we are using a self-consistent measure which represents the short-axis of the dune itself, i.e., the shortest distance across and isolated dune (schematic of Fig. 1a), most similar to the latter previous definitions but it does not depend on a recognizable inter-dune and can also be defined for superimposed dunes. True wavelength extraction using the auto-correlation would find the distance from the origin to the nearest local maxima (as has been done before for dunes[7]), however this does not allow identification of superimposed structures or the width of dunes. The representative height of a dune is found by looking at the most common range in topographic height found when looking within an area spanned by the planform wavelength of the dune.

**Mixed layer height measurements.** MLH values are found from the CALIPSO version V4-20 Level 2 aerosol layer product[20]. We identify the MLH as the lowest reported aerosol layer top height extracted from the backscatter profile at 5-km horizontal spacing over circular regions of interest (ROI) centered on each dune field. This method has been extensively evaluated in multiple cases[70-72]. The ROIs for each dune field have different diameters as to reflect the dune-field size and avoid any domains adjacent to the dune fields that have significantly different MLH dynamics. Four dune fields (Namib Sand Sea, Sinai Negev Erg, Wahiba Sands and Gran Desierto) were omitted from the CALIPSO data collection because they are coastal, where MLH dynamics are most strongly influenced by the ocean. All daytime profiles (since CALIOP is sun-synchronous) from instrument inception to the end of 2019 were collected within each ROI resulting in $n = 5,784$ MLH values. Profiles were collected for 34 dune fields such that there was no significant bias in observation times toward certain seasons for any dune field. MLH values and ROI radii are given in the Source Data file and a comparison to the Andreotti et al.[7] implicit measurement is given in Supplementary Fig. S3.

**Numerical experiment set-up & analysis.** ReSCAL[30,73], an open-source parallelizable code, is used to simulate dune growth. Details on the cellular automaton (CA) and lattice gas rules are published elsewhere extensively, notably by Narteau et al.[30]. Relative occurrence of CA transition rules that develop topography through fluid transport and avalanches are set by rate $\Lambda$ and threshold stress $\tau$ constants. We use the following values and note dune morphology and dynamics are generally insensitive to $O(1)$ changes in these parameters[74]: $\{\Lambda_E, \Lambda_C, \Lambda_D, \Lambda_G, \Lambda_T, \tau_1, \tau_2\} = \{4/t_0, 2/t_0, 0.02/t_0, 10^3/t_0, 3/t_0, 200\tau_0, 1000\tau_0\}$, for subscripts erosion ($E$), deposition ($C$), diffusion ($D$), gravity ($G$), transport ($T$), initiation (1) and saturation (2), respectively, where $\tau_0$ is the simulation stress scale.

The experiment domains are as follows. The fluid box is $750l_0$ wide and $750l_0 + \eta_0$ tall for all experiments, where $l_0$ is the grid spacing and $\eta_0$ is the initial sediment bed thickness. The sediment domain for $F_N = 1$ simulations is $750l_0$ wide and for $F_N > 1$ experiments, the sediment domain is $530l_0 \approx 750\sqrt{2}l_0$ wide so that the square sediment base can be rotated within the flow to simulate changing wind directions. The sediment domain is horizontally periodic such that sediment is conserved and is initialized as a flat bed of $\eta_0 = \{5l_0, 50l_0\}$ depending on the experiment. The fluid box is periodic in that the forcing is constant everywhere and is in equilibrium with the topography (reached offline from initialization for every change in direction before being allowed to interact with the topography). For $F_N > 1$ experiments the fluid flow direction is changed (that is, the sediment bed is rotated within the unidirectional fluid domain) at $200t_0$ intervals, where $t_0$ is the time step. All experiments are run for $10^4$ timesteps. Supplementary Movie 1 shows planform views of the experiments.

Dune geometry is found in the experiments in the following way, simplified from Methods section 'Dune Geometry Extraction' since the simulated topography is better behaved. Wavelength $x_{auto}$ is defined as double the closest distance from the origin of the autocorrelation $R_\eta$ of the elevation $\eta$ to where $R_\eta = 0$. Height $z$ is $\langle\delta\eta\rangle + \sigma_{\delta\eta}$ where $\delta\eta = \eta \star X$ as in Methods section 'Dune Geometry Extraction'. The convolution box $X$ gives the local max$\{\eta\}$ − min$\{\eta\}$ within width $x_{auto}$. Wavelength $x$ is then calibrated against manual measurement such that $x/x_{auto} = 2.21$. Dune celerity $c$ is found using the distance $d$ from the origin to the peak of a cross-correlation $\eta(t) \star \eta(t + \tau_{lag})$ such that $c = d/\tau_{lag}$. Since dunes slow down over time, $\tau_{lag}$ is chosen such that it increases linearly over time from $500t_0$ to $2 \times 10^4 t_0$ during the experiment duration to ensure no aliasing or spurious stationarity.

**Numerical experiment scaling.** The conversion from ReSCAL simulation time-steps $t_0$ and grid-spacings $l_0$ to real-world units of years and meters are not set a

priori but instead must be found by comparing real-world constants to those found through targeted numerical experiments[30,74]. This is because the scales in the simulation are clearly below the dune-scale and above the grain-scale, and hence they depend on the chosen $\Lambda$ and $\tau$ constants[74] (Methods). We note that the conversion will depend on specific details of observed real-world constants also, and these vary across dune fields; as in the second Methods section, we take representative global values for comparison.

To find $l_0$ we take the approach of Narteau et al.[30] where we match the length-scale of incipient real-world dunes $\lambda_r$ (m) to those in ReSCAL $\lambda_s/l_0$ such that $l_0 = \lambda_r/(\lambda_s/l_0)$ (m). The incipient dune wavelength has been shown in the field[2,3] to obey $\lambda_r = 2\pi L_{sat}\mathcal{A}/(\mathcal{B} - (u_{*,cr}/\overline{u_*})^2/\mu)$, where $L_{sat} = 2.2d\rho_s/\rho_f$. Hydrodynamic constants are $\mathcal{A} = 3.6$ & $\mathcal{B} = 1.9$, friction angle is $\mu = \tan(34°)$, from the ERA5 measurements we find the global mean of the critical to mean above-threshold friction velocity as $u_{*,cr}/\overline{u_*} = 0.809$, and representative values of grain diameter $d = 300\ \mu m$, $\rho_s = 2650$ kg/m³ and $\rho_f = 1.225$ kg/m³ are taken. This leaves us with a reasonable incipient dune wavelength of $\lambda_r = 34.7$ m[2,3,14]. In ReSCAL we measure the dispersion relation $\sigma(k)$ for wavenumbers $k = 2\pi/\lambda$ and find $k_{max} = k|_{\partial\sigma(k)/\partial k=0}$ as the most unstable mode and $\lambda_s = 2\pi/k_{max}$. This is done by blowing wind over sand strips of small-amplitude perturbations of wavenumbers $k$ and watching the decay or amplification of topography like $\ln(\eta) \sim \sigma t_0$. We find $\lambda_s/l_0 = 49.9$, giving $l_0 = 0.698$ m. See Supplementary Fig. S4a & c for the dispersion relationship and the experiment to measure it.

To find $t_0$ we must match sand flux magnitudes in the real-world $Q_r$ (m²/yr) and ReSCAL $Q_s t_0/l_0^2$. In the real-world we simply find the mean $Q_r = |\overrightarrow{q_r}| = 12.78$ m²/yr from the ERA5 measurements (Methods). In the simulations $Q_s = q_{s,sat}$ which can be found from the ratio $q_{s,sat}/q_{s,0,sat} = 0.171$, known for $\tau_1 = 200\tau_0$, and $q_{s,0,sat}t_0/l_0^2$ [30]. Then the timestep can be calculated as $t_0 = l_0^2(Q_s t_0/l_0^2)/Q_r$ (yr) using the $l_0$ calculated previously. To find $q_{s,0,sat}t_0/l_0^2$, we measure sand flux downwind of a non-erodible to erodible bed transition with $\tau_1 = 0\tau_0$ and all other parameters as in the numerical experiments[30]. The flux increases from the transition and saturates like $q/q_{sat} = (1 - e^{-D/L_{sat}})$ where $D$ is distance downwind of the transition[74]. We find that $q_{s,0,sat}t_0/l_0^2 = 0.25$, making $t_0 = 14.2$ h. See Supplementary Fig. S4b & d for the $q_{s,0,sat}$ calculation and the experiment to measure it.

## Data availability

The dune geometry and specific CALIPSO MLH height data generated in this study are provided in the Source Data file. The SRTM ASTER GDEM v3[16] and CALIPSO[20] data used in this study are available in the NASA Earthdata database https://earthdata.nasa.gov/. The ERA5 reanalysis[13] data used in this study are available in the Climate Data Store database https://cds.climate.copernicus.eu/. The dune-field age data used in this study are available in the INQUA Dune Atlas database https://www.dri.edu/inquadunesatlas/. Source data are provided with this paper.

## Code availability

Code to reproduce this paper can be found at https://doi.org/10.5281/zenodo.5718792.

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

## Acknowledgements

We thank Mackenzie Day and another reviewer for their constructive comments and improvements of this manuscript. We also thank Michael Kurgansky, Cyril Gadal, Clément Narteau, Olivier Rozier, Philippe Claudin, Orencio Durán & Raleigh Martin for useful discussions. Acknowledgment is made to the Donors of the American Chemical Society Petroleum Research Fund for partial support of this research through grant #61536-ND8 to D.J.J.

## Author contributions

Formal Analysis, Software, Validation, Visualization, and Writing—original draft, A.G.; Conceptualization and Methodology, D.J.J., F.F., A.G; Data Curation and Investigation, A.G., G.C., L.D.L, N.L.; Project Administration, Writing—review & editing, all authors; Resources, Funding Acquisition, Supervision, D.J.J. Conceptualization, Data Curation, Formal Analysis, Investigation, Software, Validation, Visualization, Writing—original draft, A.G.; Methodology, Project Administration, Writing—review & editing, A.G. and D.J.J.; Resources, Funding Acquisition, Supervision, D.J.J.

## Competing interests

The authors declare no competing interests.
