## [Peer Review File · Nature Communications]

What sets aeolian dune height?Reviewers' Comments:

Reviewer #1:

Remarks to the Author:

See attached PDF

Review of “What sets aeolian dune height?”

Authors: Gunn et al.

Reviewer: Mackenzie Day

Summary

This work uses data collected from a large number of dune fields on Earth to investigate the relationships between characteristic dune height and other properties of the field. Numerical modeling with ReSCAL is also employed to test end member scenarios. Overall, I think this is a valuable contribution to aeolian geoscience and represents a creative and thorough approach to an important problem. The range of fields and morphologies studied makes it one of the more comprehensive works out there for Earth dune fields, and the multiple methods all provide rigorous quantitative results. I have some suggestions below for clarity and additional references to related works, but overall I support publication with few changes.

Minor concerns

Methods: This is a jam-packed paper from a methods perspective, but the M-appendix sections each do a good job of explaining the relevant methods used. The only part I found unclear was how wavelength was calculated. It would help to add a phrase translating the mathematical explanation in M3 to what this means for physical dunes. For example, is x measured from crest to crest, trough to trough, or something else? Figure 1c (see note below) probably adds further confusion.

How was the sampling area in each field chosen? Based on data availability? With specific “clean” morphologies in mind? Looking at figure 1a/1b it seems like one might get a different result if the study area were moved 20 km to the west or north.

I think clarification is needed on the meaning/framing of the “flux directionality” measurement. After rereading the text, my understanding is that it is the resultant vector magnitude divided by the sum of all magnitudes of all vectors. Does this refer to just the sand-transporting wind vectors, or all winds measured? I don’t think the text is technically wrong anywhere, but as a reader the term “flux directionality” was confusing because 1) the actual ‘direction’ of the winds is irrelevant to this value, it’s just how variable the direction is, 2) there is no differentiation between a strong reversing wind with lots of transport and a no-transport system, and 3) a low flux directionality is associate with high variability. Again, not technically wrong, but as a reader I found this very difficult to follow as written.

The text (lines 95-100) makes strong claims about the trends observed in a fairly diffuse cloud of data. I agree that low flux directionality values are associated with high wavelengths in Figure 2C, but I don’t think there is a clear correlation there between the flux directionality and the aspect ratio, other than as limited by the fact that low aspect ratio dunes apparently don’t exist at low wavelengths. The color trend is in the x-axis direction, not the y-axis (aspect ratio) as stated in the text.

Specific comments

Figure 1: The extracted area in 1c looks incorrect. It seems like the yellow area is supposed to represent one “dune” but it clearly extends beyond a single dune. The caption notes that it is “not to scale” but why not? One could easily make x distance correct and still legible.

Line 18 and/or 147: I suggest referencing the Kocurek et al. (2010) paper on dune interactions which covers a wide range of these interaction types.

Line 68-70: Sure, the cited paper only focuses on barchan dunes, but dune pattern self-similarity for linear and transverse dune morphologies is also established in Day & Kocurek (2018). I suggest adding this reference.

Line 71: The text jumps abruptly to results here from an earlier description of methods. If the journal format allows it, I suggest adding subheadings to help with the transition.

Line 88: There are some other works on variability that are notable and not referenced here. I suggest Rubin and Hunter (1987) and/or Rubin and Ikeda (1990).

Line 147-149: Interestingly the transverse dunes seem not to produce larger heights for larger sediment supply. I wonder why. The authors could consider noting this in the text or caption as the light/dark blue lines overlap very closely and it could be easily missed.

Figure 4: It looks like the dashed/solid line legend in D should be in C. It would be helpful to have a trend line plotted for D as well. A quantitative relationship between Z and celerity would be very useful to the community, even if just fit to the part of the curve that is linear in this graph.

Line 172: “or any other hard physical constraint” – This refers to the physical parameters of the dunes themselves that are studied in this work, not “any other” physical constraint. Obviously, things like topography/landscape (mountains, the ocean, etc) will put a hard constraint on the coarsening of the dunes. This is implied later (line 201-202), but is also vague (“geologic condition”). I suggest revising to be clear about what specifically is meant here. It sounds like physical properties of the dune field vs physical properties of the surroundings?

References:

Day, Mackenzie, and Gary Kocurek. "Pattern similarity across planetary dune fields." *Geology* 46.11 (2018): 999-1002.

Kocurek, Gary, Ryan C. Ewing, and David Mohrig. "How do bedform patterns arise? New views on the role of bedform interactions within a set of boundary conditions." *Earth surface processes and landforms* 35.1 (2010): 51-63.

Rubin, David M., and Ralph E. Hunter. "Bedform alignment in directionally varying flows." *Science* 237.4812 (1987): 276-278.

Rubin, David M., and Hiroshi Ikeda. "Flume experiments on the alignment of transverse, oblique, and longitudinal dunes in directionally varying flows." *Sedimentology* 37.4 (1990): 673-684.

Reviewer #2:

Remarks to the Author:

In their manuscript *What sets aeolian dune height?*, Gunn et al. combine remote-sensing observations and numerical modeling to demonstrate that aeolian dunes continue to grow, though slowly, until limited by sediment supply or the limited duration of a stable wind regime. Nowhere does dune height appear to be limited by the height of the atmospheric boundary layer, though this has been hypothesized.

This is interesting work showing a link between dune morphology and geologic boundary conditions other than the MLH. The results presented here are well justified. Beyond this contribution, the work here will be valuable for future work towards understanding dune fields and geologic boundary conditions on other planets and moons, and likely in the ancient as well. I found the methodology to be well written and reproducible, and appreciated the test of the FFT dune geometry extraction method to measurements made by hand. For these reasons, I think this manuscript is appropriate for *Nature Communications*.

I've left only a few comments highlighting results presented here that have been reported before in the literature, but have not been cited here. These are the only minor revisions I would suggest before publication.

L23-26: This result was also published in (Swanson et al., 2017, 2019).

L67: Self similarity was documented for multiple dune types and planets/moons in (Day & Kocurek, 2018).

L90-91: A wide ranges of values for flux directionality was published in (Swanson et al., 2016).

L224: Source geometry is thought to play a role as well, and should at least be a factor in how important migration is for the expansion of the field (Ewing & Kocurek, 2010)

Authors' response to Reviewer 1:

Reviewer: Mackenzie Day

Summary: This work uses data collected from a large number of dune fields on Earth to investigate the relationships between characteristic dune height and other properties of the field. Numerical modeling with ReSCAL is also employed to test end member scenarios. Overall, I think this is a valuable contribution to aeolian geoscience and represents a creative and thorough approach to an important problem. The range of fields and morphologies studied makes it one of the more comprehensive works out there for Earth dune fields, and the multiple methods all provide rigorous quantitative results. I have some suggestions below for clarity and additional references to related works, but overall I support publication with few changes.

We thank Mackenzie Day for her thorough, constructive and supportive review. We agree with all the points raised in the review, and have made all changes as recommended; they have improved the manuscript's clarity and impact. Below we address each point made in the review directly.

Minor concerns

Methods: This is a jam-packed paper from a methods perspective, but the M-appendix sections each do a good job of explaining the relevant methods used. The only part I found unclear was how wavelength was calculated. It would help to add a phrase translating the mathematical explanation in M3 to what this means for physical dunes. For example, is x measured from crest to crest, trough to trough, or something else? Figure 1c (see note below) probably adds further confusion.

This is a fair point. The wavelength calculation algorithm is designed such that it works for any dune type, and as such, it ended up requiring a fair amount of complexity. We believe it's important to ensure that the mathematical explanation remains, but have added a few sentences at the end of the methods section M3 that parse it into something more understandable, as requested. We have also added references to Figure ED1 (which serves as a graphical explanation) where appropriate throughout the written explanation of the algorithm to improve clarity. The 'wavelength' value represents the short-axis length of a dune, which in the case where there is no obvious interdune, is essentially the trough-to-trough distance, but in the case where the dunes are clearly separated by interdune, is the size of the bedform. The point about Figure 1c is well taken and addressed below.

How was the sampling area in each field chosen? Based on data availability? With specific "clean" morphologies in mind? Looking at figure 1a/1b it seems like one might get a different result if the study area were moved 20 km to the west or north.

The sampling areas are not chosen per se, they are exhaustive. The global tiling is set by the gridding of the wind data, and *all* tiles which are majority-overlapping the manually drawn dune-field perimeters that are exclusively dune-covered (i.e. don't have ocean, other geomorphological relief in them such as mountains, and aren't just sand sheets) are included in the dune geometry analysis. We did not choose the ERA-5 grid, the dune-field perimeters were drawn using LANDSAT imagery without prior knowledge of the ERA-5 grid, and the removal of any tiles due to dune-coverage was done using LANDSAT imagery without prior knowledge of the topographic relief or geometry extraction results. The study area in Figure 1b is just an example; the paper includes the results for the adjacent tiles to the north, south and west of this example (and some further south too, see Figure 1d), and all together the 16 tiles make up the results for the Namib Sand Sea. Clearly we did not explain this method sufficiently in the manuscript and have updated the text in Methods M3 the caption of Figure 1, to show this, as well as included a file in the data and code repository (supplementary-data-GIS.kml) which shows all the tile and dune-field locations in the paper. Now Methods M3 has multiple parenthetical references to Figure ED1 and the following passages have been added on Lines 263-269 and 291-302, respectively:

"This process is done for tiles of topographic data, where the tiling is set by the gridding of the wind data such that each characteristic dune geometry found has corresponding saturated ERA-5-derived sand flux data calculated at the tile center. The tiles analyzed are those that have a majority of their area shared with a dune-field area (Methods M1) and meet the following criteria: they do not contain ocean or non-dune relief, and are not exclusively unpatterned sand sheets. These constraints leave us with 2,093 tiles to extract dune geometry from. Below we first explain the algorithm precisely, then explain how it maps onto a physical definition of dune dimensions."

I think clarification is needed on the meaning/framing of the “flux directionality” measurement. After rereading the text, my understanding is that it is the resultant vector magnitude divided by the sum of all magnitudes of all vectors. Does this refer to just the sand-transporting wind vectors, or all winds measured? I don't think the text is technically wrong anywhere, but as a reader the term “flux directionality” was confusing because 1) the actual ‘direction’ of the winds is irrelevant to this value, it's just how variable the direction is, 2) there is no differentiation between a strong reversing wind with lots of transport and a no-transport system, and 3) a low flux directionality is associated with high variability. Again, not technically wrong, but as a reader I found this very difficult to follow as written.

What we call ‘flux directionality’ is essentially ‘resultant drift potential over drift potential’ (RDP/DP) when the vectors are sand flux (not wind) defined in Methods M2. The reviewer is correct with their understanding and the three points above, and we note that there is precedent for this parameter. On point two; this is true in principle, but in practice there are no no-transport cases (i.e. the expectation in active dune fields). On point three our perspective is that low directionality should correspond to high variability, i.e. a transport system is more directional the more the transport is in one direction, and to point one, this should be true no matter which direction the system has. We are open to renaming this term but thought that ‘flux directionality’ was more succinct and descriptive than ‘resultant drift potential over drift potential’ or ‘directional variability’ as it has previously been referred to. We have rephrased and expanded the introduction of this term in the manuscript to improve clarity. Now the passage from Lines 93-96 reads: “We compute a flux directionality measure as the ratio of the magnitude of the resultant potential sand flux vector ($|\sum \vec{q}|$) over the absolute potential sand flux ($\sum |\vec{q}|$) that varies from 0 associated with net-zero flux, to 1 corresponding to unidirectional flux (Fig. 1f). This is similar to the ratio of resultant to absolute so-called ‘drift potential’ (i.e. RDP/DP).”

The text (lines 95-100) makes strong claims about the trends observed in a fairly diffuse cloud of data. I agree that low flux directionality values are associated with high wavelengths in Figure 2C, but I don't think there is a clear correlation there between the flux directionality and the aspect ratio, other than as limited by the fact that low aspect ratio dunes apparently don't exist at low wavelengths. The color trend is in the x-axis direction, not the y-axis (aspect ratio) as stated in the text.

These comments are reasonable since we did not show a statistical test and the text was slightly unclear. Lines 95-100 made two claims: lower flux directionality leads to (1) taller dunes and (2) anomalously pointy dunes. While there is plenty of scatter in the data, claim 1 is highly statistically significant: the fit parameters for a linear regression to the data fall outside the probability distribution of fit parameters if the data were shuffled (see figure below). Our point in claim 2 was that, mostly, aspect ratio decreases with wavelength, but the data that buck this trend (i.e. data in top right of Figure 1c) which have anomalously large aspect with respect to this trend, tend to have lower flux directionality. Forgetting the wind data for a second, this can be seen in morphology and geometry in Figure 1a too: mostly height increases with wavelength sublinearly, but for very large dunes there is this cluster of extremely tall dunes above that trend, and this cluster has a much higher fraction of star dunes (i.e. morphology associated with low flux directionality) in it than the rest of the population. The reviewer's point that there is a clearer trend in color with the x-axis than the y-axis is true; we do not disagree with that, but think that potentially the word ‘anomalously’ wasn't clear enough. We have heavily edited this final sentence about claim 2, and included the figure below as Extended Data Figure 4 and referenced in with claim 1 (Line 103). Now the sentence on Lines 103-105 reads: “Indeed, the previously discussed trend of decreasing aspect ratio with increasing wavelength is associated with more unidirectional sand-flux regimes, while at the largest wavelengths, the cloud of points which buck this trend and have larger-than-expected aspect ratios correspond to lower flux directionality (Fig. 2c).”

Figure ED4: Relation of flux directionality and dune height. The significance of the inverse relationship between flux directionality and dune height seen in Figure 2b is illustrated by showing how the observed fit (blue marker) parameters in a linear relationship compares to the joint density distribution (log-scale; red colors) of fit parameters for linear regression to the data when the flux directionality values are randomly shuffled with respect to the dune height values: the observed negative trend is well outside the most likely trends which would be found if the observed distributions of dune height and flux directionality were unrelated. The dashed black line indicates where the linear regression has zero slope, i.e. when changes in dune height are unrelated to changes in flux directionality (the density of fit parameters for shuffled data are bisected by this line). In the inset, the observed linear fit is shown along with the corresponding coefficient of determination R^2 .

Specific comments

Figure 1: The extracted area in 1c looks incorrect. It seems like the yellow area is supposed to represent one “dune” but it clearly extends beyond a single dune. The caption notes that it is “not to scale” but why not? One could easily make x distance correct and still legible.

We scaled the area up and moved it to the side so we could define the x and y distances while also showing the autocorrelation pattern. We have updated this panel and the caption so it’s much clearer what is going on here. Figure 1 panel c has changed slightly and the caption has small changes.

Line 18 and/or 147: I suggest referencing the Kocurek et al. (2010) paper on dune interactions which covers a wide range of these interaction types.

Done, thank you for the suggestion. Lines 20 and 157.

Line 68-70: Sure, the cited paper only focuses on barchan dunes, but dune pattern self-similarity for linear and transverse dune morphologies is also established in Day & Kocurek (2018). I suggest adding this reference.

This passage is about the dune geometry relationships (i.e. height vs wavelength), not about dislocations, but the point about dune patterns being used to infer the maturity of a dune field (and this paper) are very relevant elsewhere in this paper. We have rightfully added this reference and some text about it in the Implications section instead (Line 225), and in the Methods M3 section (Line 296).

Line 71: The text jumps abruptly to results here from an earlier description of methods. If the journal format allows it, I suggest adding subheadings to help with the transition.

We agree with the reviewer: too abrupt in hindsight. We have added subheadings here and in the Numerical Experiments section for consistency. Lines 43, 71, 124 and 152.

Line 88: There are some other works on variability that are notable and not referenced here. I suggest Rubin and Hunter (1987) and/or Rubin and Ikeda (1990).

We've added both of these important references here and also at the end of the Numerical Experiments section where relevant too, thank you for the suggestion. Now they are included on Lines 92 and 176.

Line 147-149: Interestingly the transverse dunes seem not to produce larger heights for larger sediment supply. I wonder why. The authors could consider noting this in the text or caption as the light/dark blue lines overlap very closely and it could be easily missed.

Interesting point; we have added a qualifier to this sentence. Our suspicion is that the relationship is quite similar in the unidirectional case between the high and low supply cases because the role of supply in producing interdune area is at its lowest; this is because it is the case where the most flux goes into pushing bedforms sideways instead of piling sand upwards in them (and removing it from between them). Line 158 now includes "when flux is not unidirectional."

Figure 4: It looks like the dashed/solid line legend in D should be in C. It would be helpful to have a trend line plotted for D as well. A quantitative relationship between Z and celerity would be very useful to the community, even if just fit to the part of the curve that is linear in this graph.

Yes this is correct, we have fixed the legend issue and updated the caption, thank you. We agree that a robust relationship between height and speed would be very useful, but don't feel confident to say more than that they are inversely related (back to Bagnold, which is mentioned in the manuscript already). This is because 1) there isn't a clear way to collapse the separate experiment curves onto a master curve, and 2) we can't test the veracity of the simulation bedform speeds given their size against field/lab observations in all cases (yet!). The complexity of bedform migration speeds was alluded to in the text (beginning Line 168): "higher-order effects like slip-face development and flow shielding may also reduce flux and hence migration rate as dunes become large." Figure 4c&d have legend and annotation changes and the caption ends now as "Time arrows given in (c) and (d) for clarity."

Line 172: "or any other hard physical constraint" – This refers to the physical parameters of the dunes themselves that are studied in this work, not "any other" physical constraint. Obviously, things like topography/landscape (mountains, the ocean, etc) will put a hard constraint on the coarsening of the dunes. This is implied later (line 201-202), but is also vague ("geologic condition"). I suggest revising to be clear about what specifically is meant here. It sounds like physical properties of the dune field vs physical properties of the surroundings?

The reviewer is right, this "or any other hard physical constraint" is too vague and broad, we have removed it. It was meant to mean that there is no clear-cut all-encompassing generic mechanism for creating equilibrium giant dune heights, even if that mechanism isn't a MLH capping resonance. For that reason we have included the word "generic" instead. Now Lines 180-183 read "The distilled interpretation of our findings is this: Earth's giant dunes are growing ever-slower with size, and are not limited in size by MLH generically."

References:

Day, Mackenzie, and Gary Kocurek. "Pattern similarity across planetary dune fields." *Geology* 46.11 (2018): 999-1002.

Kocurek, Gary, Ryan C. Ewing, and David Mohrig. "How do bedform patterns arise? New views on the role of bedform interactions within a set of boundary conditions." *Earth surface processes and landforms* 35.1 (2010): 51-63.

Rubin, David M., and Ralph E. Hunter. "Bedform alignment in directionally varying flows." *Science* 237.4812 (1987): 276-278.

Rubin, David M., and Hiroshi Ikeda. "Flume experiments on the alignment of transverse, oblique, and longitudinal dunes in directionally varying flows." *Sedimentology* 37.4 (1990): 673-684.

Authors' response to Reviewer 2:

In their manuscript *What sets aeolian dune height?*, Gunn et al. combine remote-sensing observations and numerical modeling to demonstrate that aeolian dunes continue to grow, though slowly, until limited by sediment supply or the limited duration of a stable wind regime. Nowhere does dune height appear to be limited by the height of the atmospheric boundary layer, though this has been hypothesized. This is interesting work showing a link between dune morphology and geologic boundary conditions other than the MLH. The results presented here are well justified. Beyond this contribution, the work here will be valuable for future work towards understanding dune fields and geologic boundary conditions on other planets and moons, and likely in the ancient as well. I found the methodology to be well written and reproducible, and appreciated the test of the FFT dune geometry extraction method to measurements made by hand. For these reasons, I think this manuscript is appropriate for *Nature Communications*. I've left only a few comments highlighting results presented here that have been reported before in the literature, but have not been cited here. These are the only minor revisions I would suggest before publication.

We thank the reviewer for their support of the manuscript. Please see below that we have incorporated all of their recommendations.

L23-26: This result was also published in (Swanson et al., 2017, 2019).

Assuming the reviewer is referring to the following two references, we agree that the 2019 paper does come to a similar conclusion and think it is relevant to cite insofar as they conclude that as dune fields become more mature, they become more sensitive to boundary conditions (as seen in dune stratigraphy). We note however that both papers specifically rely on the result that dune heights saturate to an equilibrium value, which is in conflict with this manuscript's main result. We have worked both references into the manuscript where appropriate on Lines 27, 170, 176, 221.

Swanson, T., Mohrig, D., Kocurek, G., Cardenas, B. T., & Wolinsky, M. A. (2019). Preservation of autogenic processes and allogenic forcings in set-scale Aeolian architecture I: numerical experiments. *Journal of Sedimentary Research*, 89(8), 728-740.

Swanson, T., Mohrig, D., Kocurek, G., & Liang, M. (2017). A surface model for aeolian dune topography. *Mathematical Geosciences*, 49(5), 635-655.

L67: Self similarity was documented for multiple dune types and planets/moons in (Day & Kocurek, 2018).

Day and Kocurek (2018) found self-similarity in the defect density relationship with dune spacing. The other reviewer (Day) also brought this up and we have included that reference in the manuscript on Line X (not here where we're specifically talking about the dune geometry, not the patterns). Line 225 now includes "alongside other metrics such as defect density."

L90-91: A wide ranges of values for flux directionality was published in (Swanson et al., 2016).

We think the reviewer is referring to the following paper, although we're not that certain since there aren't a wide range of flux directionality values published within it, yet it is the most relevant Swanson paper from that year. That paper is about showing that non-zero flux directionality in a predominantly unidirectional system can be seen in crestline orientation dispersion. It is an interesting result which is pertinent to the relationship between dune geometry and flux directionality, but relies on one data-point of flux directionality from a period of one year. We have referenced this paper in the recommended place. Line 97.

Swanson, T., Mohrig, D., & Kocurek, G. (2016). Aeolian dune sediment flux variability over an annual cycle of wind. *Sedimentology*, 63(6), 1753-1764.

L224: Source geometry is thought to play a role as well, and should at least be a factor in how important migration is for the expansion of the field (Ewing & Kocurek, 2010)

This is very true. We have included the reference in the prior sentence to the line the reviewer refers to, and added a few words to drive home the central message of that paper (which we had previously referenced). Line 233 now includes "*properties of the sand supply.*"

Authors' additional changes:

We mistakenly wrote that the gridded tiles had 32 km² area in a few places in the manuscript, but in fact they are 32-km wide nominally, and therefore 32² km² in area. We have corrected these instances in the manuscript in the Figures 1 and 2 and Table ED1 captions, Lines: 47, 48, 55, 61, 270.

We noticed that the final sentence in the caption of Figure 3 was not clear, it now reads: “The mean difference (red line) of the delineation between high and low β , Z , (purple line) and elevation η (grey region) for the scan constitutes one H value.”

We have added reviewer acknowledgments. Now reads: “We thank Mackenzie Day and another reviewer for their constructive comments and improvements of this manuscript.”

Andrew Gunn has changed institutions during this review process, their change in affiliation has been reflected in the manuscript. Line 5.

We mistakenly omitted a funding source in the acknowledgments for this work; this has been updated. Now reads: “Acknowledgment is made to the Donors of the American Chemical Society Petroleum Research Fund for partial support of this research through grant #61536-ND8 to D.J.J.”

We have made a DOI for the repository where the code is available that accompanies this manuscript and updated the url provided in the manuscript: doi.org/10.5281/zenodo.5718792

We have capitalized the proper nouns in bibliographic titles, which was not done automatically in our workflow, throughout the bibliography.

We have updated the Yang et al. 2019 paper bibliographic information (previously referred to as ‘in review’). It is now: Yang, X. et al. *Holocene aeolian stratigraphic sequences in the eastern portion of the desert belt (sand seas and sandy lands) in northern china and their palaeoenvironmental implications. Sci. China Earth Sci. 62, 1302–1315 (2019).*

We added the quantitative value for the representative dune migration speed we used in Figure 2e to the caption. It was defined, but not given, anywhere in the manuscript.

We have removed the references in the abstract and lowered its word count as per the formatting requirements of a *Nature Communications* Article. We removed the final sentence and part of the first sentence.

We have changed the manuscript section headings as per the formatting requirements of a *Nature Communications* Article. Lines 13, 41, 179.

We have reordered the Methods, References and End Matter statements in the manuscript, merging the Methods and Main Text bibliographies as per the formatting requirements of a *Nature Communications* Article.

Reviewers' Comments:

Reviewer #1:

Remarks to the Author:

The authors have provided thorough responses to all of my comments and the comments of the other reviewer. It appears as though all the suggested changes were incorporated and the manuscript is much improved. I see no issues with the updated version and recommend publication.